# Monoclonal Antibodies in the Treatment of Diffuse Large B-Cell Lymphoma: Moving beyond Rituximab

**DOI:** 10.3390/cancers14081917

**Published:** 2022-04-10

**Authors:** Sotirios G. Papageorgiou, Thomas P. Thomopoulos, Athanasios Liaskas, Theodoros P. Vassilakopoulos

**Affiliations:** 1Hematology Unit, Second Propaedeutic Department of Internal Medicine and Research Institute, School of Medicine, National and Kapodistrian University of Athens, University General Hospital “Attikon”, 18120 Athens, Greece; sotirispapageorgiou@hotmail.com (S.G.P.); ththomop@med.uoa.gr (T.P.T.); 2Department of Haematology and Bone Marrow Transplantation, National and Kapodistrian University of Athens, Laikon General Hospital, 11527 Athens, Greece; athliaskas@med.uoa.gr

**Keywords:** diffuse large B-cell lymphoma, DLBCL, monoclonal antibodies, antibody-drug conjugates, bispecific antibodies, CAR T cells

## Abstract

**Simple Summary:**

Diffuse large B-cell lymphoma (DLBCL) is the most common high-grade non-Hodgkin lymphoma. The current treatment combining the anti-CD20 monoclonal antibody rituximab with chemotherapy is the gold standard for frontline treatment. Although the results for patients who relapse had been disappointing, the recent research explosion on novel agents for DLBCL has reignited the hope for these patients. This review aims to summarize the exciting research results of novel therapeutic agents recently approved or under investigation for the treatment of DLBCL. We focus on novel monoclonal antibodies conjugated to cytotoxic drugs and bispecific antibodies that have shown very promising results for patients in the relapsed/refractory setting.

**Abstract:**

Although rituximab has revolutionized the treatment of diffuse large B-cell lymphoma (DLBCL), a significant proportion of patients experience refractory disease or relapse early after the end of treatment. The lack of effective treatment options in the relapsed/refractory (R/R) setting had made the prognosis of these patients dismal. The initial enthusiasm for novel anti-CD20 antibodies had been short-lived as they failed to prove their superiority to rituximab. Therefore, research has focused on developing novel agents with a unique mechanism of action. Among them, two antibody-drug conjugates, namely polatuzumab vedotin (PolaV) and loncastuximab tesirine, along with tafasitamab, an anti-CD19 bioengineered antibody, have been approved for the treatment of R/R DLBCL. Whereas PolaV has been FDA and EMA approved, EMA has not approved loncastuximab tesirine and tafasitamab yet. Results from randomized trials, as well as real-life data for PolaV have been promising. Novel agents as bispecific antibodies bridging CD3 on T-cells to CD20 have shown very promising results in clinical trials and are expected to gain approval for treatment of R/R DLBCL soon. As the therapeutic armamentarium against DLBCL is expanding, an improvement in survival of patients with R/R and higher cure rates might soon become evident.

## 1. Introduction

Diffuse large B-cell lymphoma (DLBCL) is the most common lymphoid neoplasm, accounting for ~30% of all non-Hodgkin lymphomas (NHL). The combination of cyclophosphamide, doxorubicin, vincristine and prednisone every 21 days (CHOP-21) had been the backbone of chemotherapy for DLBCL and other aggressive NHL until 2000; however, cure rates of approximately 35% had been disappointing. More intensive regimens had failed to improve this outcome [1] with the exception of the addition of etoposide to CHOP (CHOEP) and CHOP every 14 days (CHOP-14) or consolidation with high-dose therapy and autologous stem cell transplantation (HDT-ASCT) in selected patient subsets [2,3,4].

Undoubtedly, the introduction of rituximab, the first anti-CD20 monoclonal antibody approved for cancer treatment, revolutionized the treatment of lymphoid malignancies. Rituximab was approved in 1997 to treat follicular Β-NHL; however, its indications expanded to cover newly diagnosed DLBCL in combination with CHOP-21. Over two decades later, rituximab remains irreplaceable for treatment of DLBCL, but despite the definite increase in cure rates, a sizeable proportion of patients have primary refractory disease or relapse after treatment with R-CHOP. Several studies have highlighted the prognostic significance of novel prognostic biomarkers for identifying high-risk patients that could benefit from novel therapeutic modalities [5].

This review aims to summarize the newer advancements in the immunotherapy of DLBCL, including novel anti-CD20 antibodies or antibodies targeting other molecules. Novel agents recently approved or in clinical development will also be reviewed, including antibody-drug conjugates, and bispecific antibodies. The role of chimeric antigen receptor T-cell (CAR T-cell) therapy in the treatment of DLBCL will not be discussed as it is beyond the scope of this review. An overview of the discussed agents, along with their approval status is provided in Table 1 and Figure 1.

## 2. Anti-CD20 Monoclonal Antibodies

The addition of rituximab to CHOP (R-CHOP-21) dramatically improved the outcome of patients with DLBCL based on the results of several randomized trials, as described below. Novel anti-CD20 agents such as obinutuzumab and ofatumumab were meanwhile approved for B-chronic lymphocytic leukemia and follicular lymphoma and were evaluated in the setting of previously untreated or R/R DLBCL without additive value compared to rituximab.

### 2.1. Rituximab

Rituximab is a chimeric anti-CD20 IgG1 monoclonal antibody. Rituximab monotherapy is active in R/R DLBCL with overall response rates (ORR) of 37% and some complete responses (CR) [6]. The superiority of R-CHOP-21 over CHOP was clearly demonstrated in the LNH-98.5 randomized trial of 399 patients aged 60–80 years old [7,8]. At last update, the 10-year progression-free survival (PFS) was 36.5% vs. 20.1% and the 10-year OS 43.5% vs. 27.6% for R-CHOP vs CHOP respectively [8]. The superiority of rituximab-containing regimens, mainly R-CHOP-21, in younger, lower-risk patients was also demonstrated by the MabThera International Trial (MInT) Group [9]. In 823 patients with an age range of 18 to 60 years, the 6-year event-free survival (EFS) was 74.3% for the R-chemotherapy arm vs. 55.8% for the chemotherapy-alone arm with (*p* < 0.0001), while the 6-year OS was 90.1% vs. 80% respectively [9]. A substantial benefit was reported with the addition of rituximab to CHOP-21 or CHOP-14 by other randomized and real-life studies as well [10,11,12].

Although no formal randomized evidence has been generated, rituximab also greatly improved the outcomes in patients with primary mediastinal large B-cell lymphoma (PMLBCL) [13,14]. A randomized trial of 219 patients aged up to 70 years with primary central nervous system (CNS) DLBCL was also in favor of the addition of rituximab to the methotrexate-cytarabine-thiotepa regimen (MATRix). Patients treated with the MATRix combination had a complete remission rate of 49% vs. 30% of those treated with methotrexate-cytarabine plus rituximab without thiotepa vs. 23% of those treated with methotrexate-cytarabine alone, while the corresponding 2-year PFS rates were 61% vs. 46% vs. 36% (*p* = 0.051 for the comparison of the two latter groups) with similar trends in OS [15].

In the presence of rituximab, most intensive therapies are no longer better than R-CHOP-21 [16,17,18]. All this information has established R-CHOP-21 as the gold standard for the first-line treatment of DLBCL and related disorders against which any new approach should be compared.

Elderly male patients appear to benefit less from the addition of rituximab to CHOP than females, although male sex is not a risk factor in patients treated with CHOP alone [12]. Pharmacokinetic studies revealed that rituximab clearance is significantly increased in elderly males leading to a shorter elimination half-life, but this difference was not observed among younger patient subgroups [19]. Increasing exposure time or dosing of rituximab in elderly males seemed to possibly improve their outcome with acceptable toxicity [20,21].

Given the favorable effect of rituximab maintenance in follicular lymphoma, maintenance regimens were also evaluated in DLBCL. The role of maintenance rituximab after R-CHOP was investigated in a study of 632 patients who were randomly assigned to receive R-CHOP or CHOP; responders were further randomized to rituximab maintenance or observation. Maintenance of rituximab prolonged failure-free survival (FFS) in patients treated with CHOP but not in those treated with R-CHOP [10]. Several other randomized studies evaluated the maintenance of rituximab vs. observation in responders to rituximab-based immunochemotherapy [22,23]. Overall, the primary endpoints of these trials were not fulfilled, although relapses were reduced in patients who received rituximab maintenance within the AGMT13 trial. Although rituximab maintenance may benefit specific patient subgroups, such as males and those with lower-risk diseases [22,23,24], it should be noted that rituximab maintenance is not an approved indication for DLBCL. Similarly, rituximab maintenance failed to improve the outcome of patients undergoing HDT-ASCT for R/R DLBCL [25].

### 2.2. Ofatumumab

Ofatumumab is a human monoclonal IgG1κ antibody against CD20, binding to a different epitope of CD20 than rituximab and possibly leading to better complement-dependent cytotoxicity. Ofatumumab was used as monotherapy in a multicenter phase II trial of 81 patients with relapsed DLBCL that had relapsed after ASCT or were ineligible for ASCT; 95% had been previously exposed to at least one rituximab-containing regimen. The ORR to 8 weekly infusions was 11%, with a median duration of response of 9.5 months and a median PFS of 2.5 months [26]. Similarly, another small trial provided poor results, confirming the modest efficacy of ofatumumab as a single agent in the setting of rituximab-pretreated R/R DLBCL [27].

The ORCHARRD study aimed to compare ofatumumab vs. rituximab plus dexamethasone, cisplatin, and cytarabine (O-DHAP vs. R-DHAP) as salvage therapy prior to ASCT in 447 patients with R/R DLBCL. In the end, 35% of the whole population received HDT-ASCT without differences in PFS and OS between O-DHAP and R-DHAP [28].

Ofatumumab has been evaluated as a part of first-line therapy in DLBCL in combination with mini-CHOP in patients >80 years old [29] or with bendamustine in elderly patients ineligible for R-CHOP [30]. The results were promising but it is not clear how they compare with rituximab. Ofatumumab has not been approved for DLBCL.

### 2.3. Obinutuzumab

Obinutuzumab is a glycoengineered type II anti-CD20 monoclonal antibody that induces direct cell death and has better antibody-dependent cellular cytotoxicity than rituximab. It’s remarkable efficacy in follicular lymphoma and CLL led investigators to explore its role in DLBCL patients. As a single agent, obinutuzumab was associated with a 32% ORR in R/R DLBCL in the GAUGIN trial [31].

GOYA was a multicenter, open-label, randomized, phase III study that compared the efficacy of obinutuzumab plus CHOP (G-CHOP) vs. standard R-CHOP in 1418 patients with previously untreated DLBCL. The replacement of rituximab with obinutuzumab failed to improve PFS or other endpoints, irrespectively of cell-of-origin [32]; however, in a post-hoc subgroup analysis of GOYA, among the germinal center, B-cell like (GCB) group, patients with a strong-GCB molecular signature had a significant benefit by G-CHOP compared with R-CHOP. Unfortunately, this molecular classification is not available in everyday practice and this finding also requires prospective validation [33]. Most recently, GAINED, another multicenter, open-label, randomized phase III study, compared obinutuzumab plus CHOP or doxorubicin, cyclophosphamide, vindesine, bleomycin, and prednisone ACVBP) vs. R-chemotherapy in newly diagnosed young patients with DLBCL and IPI ≥ 1, applying a PET-driven treatment protocol. Both EFS and OS were similar among the two study groups, irrespectively of age, IPI, or cell-of-origin [34]. Obinutuzumab has been approved for follicular lymphoma but not for DLBCL.

## 3. Enhanced Monoclonal Antibodies

### Tafasitamab

Tafasitamab (MOR208) is an Fc-engineered IgG1 antibody targeting CD19. Fc engineering, consisting of the introduction of two amino acid substitutions (S239D and I332E), enhances the binding of the Fc domain to activating Fcγ-receptors which in turn leads to augmented antigen-dependent cell-mediated cytotoxicity (ADCC) and antigen-dependent cell-mediated phagocytosis compared with the unmodified CD19 antibody [35].

An open-label, single-arm, multicenter phase IIa trial has evaluated tafasitamab as monotherapy in the context of R/R B-NHL. Treatment comprised tafasitamab 12 mg/kg, administered as an intravenous infusion (IV) over 2 hours on days 1, 8, 15, and 22 of a 28-day cycle for two cycles. Among the 35 patients with DLBCL, the ORR and CR rates were 26% and 6%, respectively. Although the median PFS was only 2.7 months due to the low ORR, the median duration of response was remarkably high at 20.1 months over a median follow-up of 21 months. The most common any-grade adverse events were infusion-related reactions (12%) and neutropenia (12%) [36].

The single-arm phase II L-MIND trial (NCT02399085) evaluated tafasitamab in combination with lenalidomide to treat R/R DLBCL. Treatment comprised up to 12, 28-day cycles of tafasitamab, 12 mg/kg IV, q1w C1–3, and q2w C4–12 plus lenalidomide 25 mg PO d1–21, C1–12. Eighty-one patients with R/R DLBCL were recruited; among them, CR was achieved in 40.0% and PR in 17.7%, respectively, yielding an ORR of 57.5%. After a median follow-up of 12.7 months, the median PFS and OS were 12.1 and 31.6 months respectively. Notably, a remarkable duration of response of 34.6 months was noted, without significant adverse events [37]. According to updated long-term results at a follow-up exceeding 35 months, the median estimated DoR was 43.9 months, and the median PFS and OS were 11.6 and 33.5 months respectively. Notably, the complete responses appeared to be durable, as the 36-month DoCR and OS were 80.1 and 81.3%, respectively [38]. Moreover, tafasitamab with lenalidomide was more effective as second-line treatment compared to third-line or beyond (median PFS: 23.5 vs. 7.9 months; OS: 45.7 vs. 11.5 months) owing to better ORR rates; however, DoR was comparable among the two subgroups. The combination of tafasitamab with lenalidomide was effective even in other subgroups with poor prognoses. The primary refractory disease had no impact on ORR, albeit responses tended to be more short-lived, whereas refractoriness to the previous therapy had no impact on either ORR or duration of response. As expected, IPI and cell-of-origin retained their prognostic value in patients treated with tafasitamab; however, even patients with non-GCB DLBCL demonstrated promising ORR and durable responses [39]. Based on this study, tafasitamab gained FDA approval in August 2020 to treat patients with R/R DLBCL ineligible for autologous stem cell transplantation in combination with lenalidomide. A randomized, double-arm, phase II/III study of tafasitamab-bendamustine vs. rituximab-bendamustine (B-MIND, NCT02763319) for patients with R/R DLBCL, ineligible for ASCT is ongoing, whereas other studies for newly diagnosed patients are also ongoing (Table 2).

## 4. Checkpoint Inhibitors

The PD-1 pathway is an immune checkpoint that prevents T-cell activation and proliferation, promoting immune tolerance. PD-1 ligands (PD-L1, PD-L2) are commonly expressed on the surface of neoplastic cells due to cytogenetic abnormalities in the 9p24 locus favoring escape from immune surveillance, especially in classical Hodgkin lymphoma (cHL) and PMLBCL. Immune checkpoint inhibitors have already produced impressive results in patients with HL [51,52,53]. Nivolumab and pembrolizumab have shown impressive ORR and durable responses, as suggested by the 5-year follow-up results of CheckMate 205 and KEYNOTE-087 studies, respectively [54,55]. As PD-L1 expression in DLBCL has been associated with the non-GCB subtype and inferior overall survival, several studies aimed to investigate the role of the anti–PD-1 monoclonal antibodies and other checkpoint inhibitors in DLBCL.

### 4.1. Nivolumab

Overall, nivolumab has not produced satisfactory results in DLBCL. In a phase Ib trial including 11 patients with R/R DLBCL treated with nivolumab, the ORR was 36% [56]. The efficacy of nivolumab was also evaluated in a multicenter, single-arm, open-label phase II study of 121 patients with R/R DLCBL who had relapsed after ASCT or were ineligible for ASCT and had received ≥2 prior treatment regimens. At a median follow-up of 9 months for post-ASCT patients and 6 months for ASCT-ineligible patients, ORR was 10% and 3%, the median PFS was 1.9 and 1.4 months, respectively, probably due to the low incidence of 9p24.1 copy gain and amplification and low membranous PD-L1 expression, as demonstrated by FISH analysis and immunohistochemistry [57]. The role of nivolumab in PMLBCL is discussed below in the section on antibody-drug conjugates.

### 4.2. Pembrolizumab

Pembrolizumab was evaluated in combination with R-CHOP in a study of 27 patients with previously untreated DLBCL, including 3 patients with follicular lymphoma. ORR was 90% (CR 77%) and 3-year PFS and OS were 83% and 86%, respectively, with similar toxicity to standard R-CHOP. IPI, bulky disease, and absence of PD-L1 expression were related to inferior OS. Of note, 83% of the cases were PD-L1 positive by IHC, even though neither gains nor implications of 9p24.1 were observed [41,58]. Pembrolizumab was also used in the post-ASCT setting as a consolidation treatment in a phase 2 single-arm of 29 patients. The 18-month PFS was 59%, which was below the protocol-specified primary objective and thus, the establishment of a larger confirmatory study cannot be supported [59].

However, pembrolizumab appears to have remarkable and durable efficacy against PMLBCL. KEYNOTE-013 (21 patients) and KEYNOTE-170 (53 patients) were multicenter, multicohort open-label phase 1b and II trials, respectively, which evaluated the efficacy, safety, and tolerability of pembrolizumab in heavily pretreated patients with R/R PMLBCL. The ORR was 45–48%, while the median duration of response was not reached in either study [60]. A longer follow-up of the KEYNOTE-170 study provided optimistic results as 76% of patients had a response duration > 36 months. Median PFS and OS were 5.5 and 22.3 months. The final analysis of KEYNOTE-170 presented at the 2021 ASH annual meeting demonstrated prolonged DoR [median not reached (NR)] during a median follow-up of 48.7 months. Based on these results, pembrolizumab has gained FDA approval for treatment-refractory PMLBCL or for patients who have relapsed after at least two prior treatment lines. Overall, pembrolizumab emerges as a highly promising treatment option for patients with R/R PMLBCL but has not been evaluated as Pembro-R-CHOP in previously untreated patients.

### 4.3. Magrolimab

Magrolimab (Hu5F9-G4) is a new humanized IgG4 monoclonal antibody targeting CD47. CD47 is a transmembrane protein ubiquitously expressed on normal tissues. When bound to signal regulatory protein alpha (SIRPa), an inhibitory receptor on macrophages and dendritic cells, CD47 sends a ‘don’t eat me’ signal. Cancer cells overexpress CD47 as a mechanism to evade phagocytosis, making it an appealing target for monoclonal antibodies [61]. CD47 blockade promotes phagocytosis of neoplastic cells by macrophages and tumor antigen-presentation by phagocytes to T cells. Magrolimab was administered in combination with rituximab in a phase Ib study of 15 patients with R/R DLBCL. ORR was 40% and 33% of patients achieved CR. Of note, 95% of DLBCL patients had been refractory to rituximab-based regimens, suggesting that magrolimab could potentially lead to the reversal of previous rituximab refractoriness. Responses were rather durable as the median duration of response was not reached at a median follow-up period of 6.2 months. Rituximab and magrolimab might have a synergistic anti-tumor effect by combining the phagocytic effect of magrolimab with rituximab-induced antibody-dependent cellular phagocytosis [62].

### 4.4. Other Checkpoint Inhibitors

Atezolizumab, durvalumab, and avelumab are novel PD-L1 inhibitors that have been approved for the treatment of several solid tumors. Their efficacy in the frontline setting in patients with DLBCL is currently being investigated and preliminary results of these studies are presented in Table 2. However, their results in R/R DLBCL have not been promising; Atezolizumab failed to demonstrate any survival benefit when combined with obinutuzumab or tazemetostat and durvalumab has failed to show any additive efficacy in combination with ibrutinib, BR, or R-lenalidomide.

Although the efficacy of checkpoint inhibitors in R/R DLBCL has been limited in general, accumulating data suggest that these agents may prevent disease relapse or overcome resistance in a considerable number of patients following CAR T-cell therapy [63]; therefore, they merit consideration in the post CAR T setting.

## 5. Radioimmunotherapy

Radioimmunotherapy (RIT) is a form of targeted therapy that uses monoclonal antibodies conjugated to radioisotopes, thus delivering irradiation specifically to tumor cells.

^90^Y ibritumomab tiuxetan is a murine IgG1 monoclonal antibody against CD20 bounded to ^90^Yttrium, approved for the treatment of indolent lymphomas, but research in aggressive NHL has been limited. In DLBCL, it was first evaluated as a post CHOP consolidation treatment in a phase II study of 20 patients and demonstrated a CR rate of 95%. Notably, four out of the five patients initially in PR following chemotherapy alone achieved CR after RIT consolidation and 2-year PFS was estimated to be 75% [64]. Another study included 55 high-risk untreated patients > 60 years old, who were treated with a shorter course of R-CHOP (4 cycles) and then received ^90^Y ibritumomab tiuxetan. The CR rate was 73% and the 2-year disease-free survival (DFS) and OS were 85% and 86%, respectively [65]. In a recent phase II study of 20 patients with DLBCL, R-CHOP-14 was followed by consolidation with one course of ^90^Y ibritumomab tiuxetan. Notably, the CR rate was increased to 95% after consolidation compared to 75% at the end of immunochemotherapy, as 5 out of 7 patients in PR achieved CR [66].

Another anti-CD20 antibody, tositumomab bound to ^131^Iodine, although FDA- approved for R/R low-grade, follicular, or transformed NHL, has been discontinued due to lack of demand. Nevertheless, results of a phase II study on 56 patients with DLBCL had demonstrated 2-year PFS and OS of 69% and 77%, respectively, when administered as consolidation after standard treatment with R-CHOP, showing the minimal additive value of RIT consolidation [67].

^90^Yttrium has also been combined with epratuzumab, an anti-CD22 humanized antibody, in a prospective, single-group phase II study of 71 patients. Patients were 60–80 years old and had previously untreated bulky or advanced disease and were not eligible for ASCT in case of primary refractory disease or relapse. The treatment consisted of 6 cycles of R-CHOP-14 followed, 6–8 weeks later, by two weekly infusions of ⁹⁰Y-epratuzumab. Notably, following RIT consolidation, 68% of patients achieved CR, including 12 patients that were in PR following R-CHOP [68].

## 6. Antibody-Drug Conjugates

Antibody–drug conjugates (ADCs) are immunoconjugates comprised of a monoclonal antibody linked to a cytotoxic drug (known as the payload) via a chemical linker. The ADC is designed to selectively deliver an ultra-toxic payload directly to the target cancer cells. Upon binding to the corresponding antigen on the surface of tumor cells, the ADC/antigen complex is internalized and then the payloads are released, leading to cytotoxicity and cell death [69]. Currently, four ADCs are approved for the treatment of lymphoid malignancies, namely brentuximab vedotin, polatuzumab vedotin, loncastuximab tesirine, and inotuzumab ozogamicin, whereas several other molecules are under investigation in clinical trials. Details on the structure of ADCs are provided in Table 3.

### 6.1. Commercially Available Antibody-Drug Conjugates

#### 6.1.1. Brentuximab Vedotin

Brentuximab vedotin (BV) is an ADC targeting CD30. Since 2011, BV has gained FDA and EMA approval to treat various subsets of R/R and newly diagnosed cHL and T-cell lymphomas, including anaplastic and cutaneous T-cell subtypes.

CD30 is variably expressed in a subset of high-grade B-cell lymphomas, including PMLBCL and DLBCL. In a large series, CD30 was expressed in ~14% of DLBCL patients [70]. Thus, BV has been evaluated for the treatment of both these entities in clinical trials summarized in Table 4. Monotherapy with BV has demonstrated moderate efficacy for R/R DLBCL with ORR not exceeding 50% and rather short PFS. Notably, the observed responses were irrespective of CD30 expression status [71,72,73]. In contrast to what was expected in PMLBCL, the ORR was rather disappointing in a recent study, as only two out of 15 enrolled patients responded, both with PR [74]. Although BV has not been granted approval by the regulatory authorities, the occurrence of responses and the rather prolonged DoR make BV a potential bridging therapy to other more radical approaches, such as allogeneic SCT or CAR T-cell therapy in DLBCL.

Whereas the aforementioned studies evaluated the efficacy of BV monotherapy in R/R DLBCL, the combination of BV with lenalidomide was evaluated in the same setting [75]. In this phase I trial, 37 DLBCL patients were enrolled; among them, 22 patients were CD30-negative. The ORR was 57%, with 13 patients achieving CR. Notably, 7 of 13 CR patients were CD30-negative. Another phase II trial evaluating the combination of BV with bendamustine and rituximab was prematurely terminated by the sponsor based on portfolio prioritization (NCT02594163).

More importantly, CheckMate 436 investigated the role of BV in combination with nivolumab in 30 patients with R/R PMLBCL. The ORR was 73% with 37% CRs. At a median follow-up of 33.7 months, the median DOR was 31.6 months, the median duration of CR was not reached, and the median PFS was 26.0 months [76,77]. However, given the rather disappointing results of BV monotherapy in PMLBCL, the additive value of BV on nivolumab is not clear.

BV has also been studied as a frontline treatment of DLBCL (Table 2). In the phase II clinical trial, BV was combined with standard R-CHOP in 51 high-risk DLBCL patients, defined as IPI ≥ 3 or age-adjusted IPI ≥ 2. The 18-month PFS for the 25 CD30-positive and the 24 CD30-negative patients) was 79% versus 58%, respectively. Based on the high rate of grade ≥ 3 peripheral neuropathy in patients treated with BV + R-CHOP, vincristine was omitted from the combination in a subsequent part of this study, and patients were homogeneously treated with BV + R-CHP. Among 11 evaluable DLBCL patients, ORR was 91% with 82% CRs [44]. The combination was also evaluated in a population of high-grade B-NHL with excellent results, as shown in Table 2 [45]. Preliminary findings of a study focusing on the elderly (>75 years) with newly-diagnosed DLBCL were promising, as shown in Table 2 [46].

In conclusion, BV might be a relatively appealing bridging option in heavily pretreated DLBCL patients and appears feasible and promising when combined with standard immunochemotherapy in the frontline setting. However, randomized evidence is needed to define the additive efficacy of BV to CHP, particularly in the setting of CD30- patients, as standard R-CHOP remains highly effective in DLBCL and even in PMLBCL [91]. Toxicities, particularly peripheral neuropathy, should also be considered [92].

#### 6.1.2. Inotuzumab Ozogamicin

Inotuzumab ozogamicin (InO) is an ADC targeting CD22, solely expressed on the surface of B lymphocytes [93]. InO was approved by the FDA and the EMA in 2017 to treat R/R B-cell precursor acute lymphoblastic leukemia (ALL) [94], whereas several trials have investigated its efficacy in the treatment of B-NHL, as summarized in Table 4.

Early-phase studies employing InO monotherapy or the combination of InO with rituximab have yielded conflicting results [78,79,80]. A large phase III randomized trial failed to show any superiority of R-InO to conventional immunochemotherapy (IC). In this study, 338 patients with R/R aggressive B-NHL (91% DLBCL) were randomized to receive either 6 cycles of R-InO or 6 cycles of rituximab-bendamustine (BR) or rituximab-gemcitabine. The ORR was 41% for R-InO compared to 44% for IC, with 16% CRs in both groups, while PFS and OS were almost indistinguishable. Notably, an increased risk for adverse events in the arm of R-InO was noted, especially in terms of severe thrombocytopenia and hepatic adverse events, namely hyperbilirubinemia and veno-occlusive liver disease (VOD) [81].

InO has also been studied in combination with conventional immunochemotherapy. In a phase I trial, InO in combination with R-CVP yielded an ORR of 57% (CR: 10%) among 21 patients with R/R DLBCL [82], whereas another study investigating the efficacy of InO in combination with rituximab, gemcitabine, dexamethasone, and cisplatin (R-GDP), yielded comparable results (ORR: 44%, CR: 21%) [83]. Notably, these response rates converge to those of chemoimmunotherapy alone, questioning the additive efficacy of InO to standard chemoimmunotherapy.

In conclusion, InO has demonstrated relatively moderate efficacy in DLBCL, whereas the increased risk of VOD associated with its use raises major concerns. High-grade hematologic toxicities, neutropenia, febrile neutropenia, and thrombocytopenia have also been observed with higher frequency in patients treated with InO across clinical trials, irrespective of the treatment indication [95].

#### 6.1.3. Polatuzumab Vedotin

Polatuzumab vedotin (PolaV) is an ADC targeting CD79b, a component of the B-cell receptor, invariably expressed in B-NHL; details on its structure and mechanism of action are presented in Table 3. PolaV was granted accelerated approval by the FDA in June 2019, in combination with bendamustine and rituximab (BR) for the treatment of R/R DLBCL after at least two prior therapies, whereas EMA approved PolaV in combination with BR for the treatment of adult patients with R/R DLBCL who are not candidates for hematopoietic stem cell transplant even as a second-line option.

PolaV has demonstrated considerable single-agent activity in early phase studies. Notably, in the randomized phase II trial GO29365 (NCT02257567), PolaV was combined with bendamustine and rituximab (BR) or obinutuzumab (BO) for patients with R/R DLBCL ineligible for ASCT; in the phase II part of the trial, 40 patients received PolaV-BR, and 40 BR only. The addition of PolaV significantly improved ORR (45% vs. 18%), CR (40% vs. 17.5%), OS (median 12.4 vs. 4.7 months), and PFS (median 9.5 vs. 3.7 months). Regarding tolerability, PolaV-BR was associated with a higher rate of grade 3–4 cytopenias but no higher risk of infections or need for transfusion. Notably, peripheral neuropathy in the PolaV arm was low-grade and reversible [96]. The extension of this study on 106 patients who received PolaV-BR demonstrated the best overall response (BOR) of 56.6%, whereas ORR was 41.5% with 38.7% CRs. The median duration of response was 9.5 months, whereas PFS and OS were 6.6 and 12.5 months, respectively. Interestingly, a subgroup analysis showed inferior PFS and median duration of response in patients with primary refractory disease, those with refractory to last treatment, and those who had received more than one prior therapy [88].

Data on the efficacy and safety of PolaV-BR in the real-life setting are rapidly accumulating, as summarized in Table 5. In general, real-world data show comparable response rates; however, PFS and OS are inferior to those demonstrated in the GO29365 trial. This discrepancy might be attributed to the inclusion of more heavily pretreated patients with poor performance status that would be excluded from a clinical trial. Moreover, more patients in the real-world setting had failed prior to ASCT or CAR T-cell therapy, consisting of a more hard-to-treat cohort. Nonetheless, the promising response rates and the durability of the responses in line with the clinical trial confirm that PolaV-BR might be used as a bridge to HSCT or CAR T-cell therapy. Indeed, in a recent study by Liebers et al., PolaV containing regimens were used as a bridging therapy to allo-HSCT or CAR T-cell therapy. The latter was successfully performed in 28 of 41 patients with a 6-month and 12-month OS of 77.9% and 58.5%, respectively. In contrast, the 6-month OS of patients who failed to proceed to CAR T was only 22% [97]. Data on the use of PolaV after CAR T-cell therapy have been more limited. A recent retrospective study on 44 patients in relapse after CAR T demonstrated an ORR of 45%, with only 14% CRs and a median PFS of 9 weeks [98].

Ongoing trials are investigating PolaV in combination with gemcitabine and oxaliplatin (NCT04182204), obinutuzumab and atezolizumab (NCT02729896), rituximab and venetoclax (NCT02611323), as well as, rituximab and lenalidomide (NCT02600897).

PolaV has been assessed in the frontline setting, in combination with CHP and either rituximab or obinutuzumab, as shown in Table 2. POLARIX (NCT03274492) is the first phase III trial introducing PolaV in the treatment of newly diagnosed DLBCL patients, in which 440 patients were randomized to PolaV-R-CHP whereas 439 received conventional R-CHOP. All patients received 6 cycles of the respective regimen followed by rituximab monotherapy for two additional cycles. After a median follow-up of 28.2 months, patients treated with PolaV-R-CHP had significantly longer PFS (76.7 vs. 70.2% at 2 years, *p*: 0.02), whereas no OS difference was demonstrated. Notably, although CR rates were comparable among the two groups, the statistically significant differences in DFS suggest more durable responses achieved with PolaV-R-CHP. Subgroups that might benefit more from PolaV-R-CHP include patients older than 60 years, those with IPI scores ≥ 3, and those with the activated B-cell–like subtype. Most importantly, the safety profile was comparable between the two groups [48].

#### 6.1.4. Loncastuximab Tesirine

Loncastuximab tesirine (ADCT-402) is an ADC targeting CD19. Structural details can be found in Table 3. A phase I trial in 139 DLBCL patients demonstrated 42.3% ORR with 23.4% CR and a median duration of response (mDOR) of 4.5 months. PFS and OS were 2.8 and 7.5 months respectively. The most common serious adverse events included cytopenias and elevation in gamma-glutamyl transferase (GGT), photosensitivity, pleural/pericardial effusions, and edemas [89].

A phase II trial (LOTIS-2, NCT03589469) of 145 R/R DLBCL demonstrated an ORR of 48.3% and a CR rate of 24.1%, with a median duration of response of 12.6 months. Median PFS was 4.9 months and the median OS was 9.9 months. Notably, loncastuximab tesirine remained an effective treatment option in patients relapsing after ASCT or CAR T-cell therapy with ORR of 58.3% and 46.2%, respectively. A subgroup analysis of 11 double-hit (DH) or triple-hit (TH) lymphomas within the LOTIS-2 trial demonstrated an ORR of 45.5%, with all responders achieving CR with a remarkable duration exceeding one year. Adverse events associated with the PBD payload, such as photosensitivity and edemas, were mild and easily manageable with supportive measures [90]. Based on these results, loncastuximab tesirine gained FDA approval in April 2021 for treatment of R/R DLBCL after at least two lines of treatment. A phase III randomized trial is currently evaluating the efficacy of loncastuximab with rituximab vs. R-gemcitabine and oxaliplatin (NCT04384484).

Loncastuximab tesirine combinations with durvalumab (NCT03685344) and ibrutinib (NCT03684694) are currently evaluated in two ongoing phase I trials; interim results of the latter study on 28 R/R DLBCL patients are remarkable as ORR reached 77.3% and 47.7% of patients achieved CR. The most common grade ≥ 3 adverse events were anemia, thrombocytopenia, and neutropenia in 8.8%, 5.9%, and 5.9% of patients, respectively.

Initial concerns on the efficacy of CAR T cell therapy following treatment with CD19 antibodies seem to be disputed. A study of 14 DLBCL patients relapsing after treatment with loncastuximab tesirine demonstrated no loss of CD19 expression. Most importantly, both efficacy and safety of CAR T cells were comparable to published data [104]

### 6.2. Investigational Antibody-Drug Conjugates

#### 6.2.1. Other Anti-CD19

Apart from loncastuximab tesirine, two other anti-CD19 ADC have been evaluated in clinical trials for DLBCL patients, namely denintuzumab mafodotin and coltuximab ravtansine, but their development has been terminated. The chemical structures and properties of these ADCs are summarized in Table 3. A phase I trial (NCT01786135) of denintuzumab mafodotin (SGN-CD19A) in patients with R/R DLBCL revealed a 33% ORR (22% CR) and a median duration of response of 40 weeks; however, a high rate of ocular toxicity was observed, with superficial microcystic keratopathy diagnosed in 84% of patients [105]. Two-phase II trials evaluating the combination of SGN-CD19A and other regimens (NCT02855359: R-CHP or R-CHOP and NCT02592876: R-ICE [rituximab, ifosfamide, carboplatin, and etoposide]) have been terminated by the sponsor based on portfolio prioritization. Regarding coltuximab ravtansine (SAR3419), a phase II trial reported moderate ORR with considerable hematologic toxicity [106], whereas another phase II trial investigating the combination of coltuximab ravtansine with rituximab among 45 R/R DLBCL patients, yielded an ORR of 31% failing to meet its primary outcome of ORR > 40% [107]. To our knowledge, there are no ongoing trials utilizing coltuximab ravtansine as monotherapy or in combination with other agents.

#### 6.2.2. Other Anti-CD22

Apart from InO, another anti-CD22 antibody-drug conjugate had been investigated in clinical trials, namely pinatuzumab vedotin (PiV, DCDT2980S). In a phase I trial of 25 R/R DLBCL patients, monotherapy with PiV yielded a modest ORR of 25% [108]. Subsequently, PiV was investigated in combination with rituximab in the context of the ROMULUS phase II trial. Although ORR was relatively high (57%) and comparable to PolaV, the median duration of response was significantly shorter for PiV (6.2 vs. 13.4 months). On this ground, PolaV was selected over PiV for further clinical development [85].

#### 6.2.3. Anti-CD25

CD25 is the α subunit of the IL-2 receptor (IL-2Rα) involved in the signal transduction for the growth and survival of immune cells. Camidanlumab tesirine, an ADC targeting CD25, had been investigated in a phase I trial, also enrolling 18 R/R B-NHL patients, yielding a moderate ORR of 30.8% with a disappointing CR of 15.4% [109]. It should be noted that patients with heavily pretreated R/R HL have shown high ORR when treated with this agent; therefore, camidanlumab tesirine might represent a promising therapeutic agent for cHL.

#### 6.2.4. Anti-CD37

CD37 is a member of the tetraspanin family found on pre-B and mature B cells, which mediates apoptotic cell signaling and is important for B and T-cell interactions. Naratuximab emtansine (IMGN529) is an anti-CD37 ADC that has been investigated in a dose-escalation phase I trial of 24 R/R DLBCL patients demonstrating poor ORR of only 22.2% [110]. Preclinical data supporting that rituximab may augment the anti-tumor activity of naratuximab emtansine [111] have triggered a phase II trial of naratuximab emtansine combined with rituximab in patients with R/R NHL (NCT02564744). Preliminary results on 100 patients, including 80 patients with DLBCL, have demonstrated ORR of 50%, including 43% CRs, with mDOR not being reached during a median follow-up of 15 months [112]. AGS67E, another anti-CD37 antibody-drug conjugate, has been evaluated in the setting of R/R DLBCL, albeit the results of a small phase I trial have not been promising [113].

#### 6.2.5. Anti-CD70

CD70, also known as CD27L, is a member of the TNF receptor superfamily. The interaction between CD70 and CD27 is critical for B cell activation, T helper 1 (Th1)/Th2 switching, and cell differentiation. CD70 is highly expressed in several malignancies, including NHL. Three molecules targeting CD70 have been investigated in phase I clinical trials. Their limited efficacy, along with their toxicity profile, hindered further development [114,115,116].

#### 6.2.6. Anti-CD79b

Except for PolaV, another anti-79b antibody-drug conjugate is currently under development. Iladatuzumab vedotin (DCDS0780A), utilizing a novel THIOMAB technology for payload conjugation in a fixed drug-to-antibody (DAR) of 2:1, has been investigated in a phase I trial both as monotherapy as well as in combination with rituximab. Fixed DAR allowed for administration of higher doses (≥2.4 mg/kg) of the ADC in comparison to PolaV, resulting in a considerable ORR of 59% with 41% CR. Notably, patients treated in combination with rituximab achieved durable responses with median DoR and PFS of 15.7 and 17.3 months, respectively; however, increased doses resulted in a different toxicity profile, predominated by ocular toxicity, often leading to dose reductions and or delays [117].

#### 6.2.7. Anti-ROR1

ROR1 is a transmembrane protein aberrantly expressed in various malignancies, exerting its oncogenic action by activation of the WNT signaling pathway. Zilovertamab vedotin (MK-2140) is an ADC targeting ROR1. Preliminary results of a phase I study on 13 R/R DLBCL patients yielded an ORR of 38.5% with 3 patients achieving CR with a manageable safety profile, leading to a currently ongoing phase II trial (NCT05144841) [118].

## 7. Bispecific Antibodies

Bispecific antibodies (BsAbs) are engineered proteins that can simultaneously recognize two different antigens or epitopes. This dual-specificity offers several advantages; notably, BsAbs can redirect immune effector cells (cytotoxic T-cells or NK-cells) to the proximity of tumor cells, thus enhancing cytotoxicity, may allow the simultaneous blocking of two different pathways with unique or overlapping pathogenetic functions and may increase binding specificity by interacting with two different cell-surface antigens [119]. BsAbs can be divided into two broad categories: immunoglobulin G (IgG)-like molecules and non-IgG-like molecules. IgG-like BsAbs retain the traditional monoclonal antibody structure of two Fab arms and one Fc region, except that the two Fab sites bind different epitopes. The presence of the Fc region allows for the Fc-mediated effector functions such as antibody-dependent cell-mediated cytotoxicity (ADCC), complement-dependent cytotoxicity (CDC), and antibody-dependent cellular phagocytosis (ADCP), and prolongs the BsAbs serum half-lives. On the other hand, Non-IgG-like BsAbs consist of single-chain variable fragments (scFvs) generated by fusing variable domains of the IgG heavy chain (VH) and light chain (VL) through a flexible polypeptide linker. The structure of BsAbs is depicted in Figure 2.

Construction of BsAbs requires various steps. Most commonly, two pre-established hybridomas expressing two different monoclonal antibodies are fused to generate a quadroma that expresses both parental heavy chains (HCs) and light chains (LCs). However, the random assembly of different heavy and light chains can produce ten different products. The functional bispecific antibody (Triomab) represents only a small proportion of the final product and is difficult to isolate and purify. To increase the output of BsAb production, several methods have been applied. In the Knobs-into-holes approach (amino acids with small side chains on one CH3 region are replaced with larger ones to generate a knob, and large side chains on the corresponding CH3 domain are replaced with smaller ones to generate a hole. The modifications favor the desirable heterodimerization of heavy chains. Mispairing of light chains can be solved by the CrossMab technology. In this approach, CH1 and CL are switched to construct a modified heavy and light change. As a result, the correct pairing between the unmodified and modified chains is augmented. Another approach to circumvent light chain mispairing is to use a common light chain that interacts with two different heavy chains. Other methods include the construction of a BsAb with an XmAb configuration, where a Fab domain is replaced by a single-chain variable fragment (ScFv); the asymmetric nature of the product favors correct assembly. Fusion of ScFv to the Fc domain and introduction of a second Fab or a single variable domain to the variable domains of a monospecific antibody with linker results in high output production with no homodimerization and low rates of mispairing [120]. In a different approach, named controlled Fab-arm exchange (cFAE), two parental antibodies containing matching mutations in the CH3 mutation are expressed separately and mixed in vitro. The recombinant end-product, termed DuoBody, demonstrates high purity with low levels of homodimer by-products [121].

Construction of non-IgG-like formats is, in general, more straightforward. BiTE (bispecific T-cell engager), consisting of a single polypeptide chain containing the four variable domains of heavy and light chains, is expressed by a single vector in which the corresponding genome has been integrated. The DART (Dual-affinity re-targeting antibody) format consists of two chains, both containing the variable regions of heavy chain and light chain from different antibodies, stabilized by a C-terminal disulfide bridge. Both formats have a particularly small size and, therefore, a very short half-life. Methods to circumvent rapid clearance include Tandem diabodies which consist of two pairs of HC and LC variable domains in a single polypeptide chain, conjugation of LCs to albumin, or fusion of Fab region using the dock-and-lock (DNL) method in which the specific protein/protein interactions between the regulatory subunit of cAMP-dependent protein kinase (PKA) and the anchoring domains of A-kinase anchor proteins (AKAPs) are used to create a trivalent stable trimer with high molecular weight [122].

### 7.1. Targeting CD20

Several CD3xCD20 BsAbs are currently under development to treat DLBCL; among them, glofitamab, mosunetuzumab, epcoritamab, and odronextamab have produced more mature results. Although these BsAbs target the same epitopes, they present several differences in terms of structure, route of administration, and treatment schedule. Table 6 illustrates the differences between the available CD3xCD20 BsAbs. Clinical trials evaluating BsAbs in DLBCL are summarized in Table 7. A head-to-head comparison of trials focusing on CD3xCD20 BsABs in terms of inclusion criteria, patients’ characteristics, efficacy, and safety is provided in Table 8.

#### 7.1.1. Glofitamab

Glofitamab (RO7082859 or RG6026) is a novel IgG-like CD3xCD20 Crossmab (2:1) BsAb; the molecule comprises two CD20 binding Fabs (derived from the Type II CD20 IgG1 obinutuzumab), one CD3 binding Fab fused to one of the CD20 Fabs via a short flexible linker, and an engineered, heterodimeric Fc region with completely abolished binding to FcγRs and C1q, decreasing its clearance and therefore extending its half-life (Figure 2d).

In the NP30179 phase I study, 258 patients with R/R NHL, including 183 patients with aggressive NHL (98 with de novo DLBCL), were treated with glofitamab in a fixed dosing or a step-up dosing scheme; as per protocol, all patients received pretreatment with a single infusion of obinutuzumab to reduce the potential risk of cytokine release syndrome (CRS). Overall, among the subgroup of aggressive NHL, ORR, and CR rates were 53.7% and 39.4%, respectively. mDOR was 29.4 months, whereas the median duration of CR had not been reached, with 72.5% still in CR at a median follow-up of 13 months (maximum 36 months). Median PFS was 2.9 months with a plateau of approximately 24% from 8 months onwards. Notably, in a sub-cohort of 14 patients treated with the recommended phase II dose (RP2D) of 2.5/10/30 mg, ORR and CR rate was 78.6 and 71.4%, respectively. Regarding safety, neutropenia was the most common grade ≥ 3 adverse event, noted in 25.1% of patients. CRS of grade ≥ 2 occurred in 71.7% at fixed doses of 10–25 mg; as expected, the step-up approach decreased the risk of CRS grade ≥ 2 (63.5%), which was confined to the first two cycles, whereas no grade 3 neurologic events were noted [123].

Glofitamab is also investigated in combination with gemcitabine and oxaliplatin (NCT04313608) and atezolizumab or PolaV (NCT03533283). Preliminary results of the combination of glofitamab with PolaV were presented to the 2021 ASH annual meeting, demonstrating a notable ORR of 73% with 51.5%, albeit the short follow-up precludes any conclusions on DoR or survival estimates. Notably, no CRS or neurological toxicity of grade ≥ 3 was documented. Another phase I trial examines the feasibility of glofitamab administration in combination with the standard R-CHOP regimen in the frontline treatment of DLBCL patients (NCT03467373). Early results have shown the feasibility of the combination, as minimal CRS and neurotoxicity were documented.

#### 7.1.2. Epcoritamab

Epcoritamab (GEN3013) is a subcutaneously administered CD3xCD20 DuoBody (Figure 2j). Preliminary results of an ongoing phase I/II trial on 68 patients with R/R B-NHL, including 46 patients with DLBCL, have been recently published. Among 33 patients evaluable for response, ORR was 68% for patients treated with 12–60 mg of epcoritamab; however, in the subgroup of 11 patients treated with ≥48 mg -the RP2D- demonstrated a remarkable ORR of 91% with a CR rate of 55%. The median time to respond was relatively short, estimated at 1.4 months. Notably, during a median follow-up of 10.2 months, all patients who achieved CR remained in remission; median PFS was 9.1 months for patients treated with doses ≥ 12 mg and not reached for patients who received 48 mg. Moreover, a trend towards a deepened response was noted, as five patients with initially, PR eventually achieved CR. Finally, this agent was effective in all four patients who had previously undergone CAR T-cell therapy, with 2 patients achieving CR and 2 PR. Regarding safety, most adverse events were grade ≤ 2; no grade 3 CRS was noted, whereas grade ≥ 3 neurotoxicities were observed in 5.1% of patients, being always transient [124].

A phase III trial is currently ongoing (NCT04628494) that aims to compare the efficacy of epcoritamab vs. conventional immunochemotherapy, namely BR or R-GemOx (rituximab, gemcitabine, and oxaliplatin) in R/R B-NHL. A phase I/II of epcoritamab in combination with R-CHOP in newly diagnosed patients with DLBCL is also currently ongoing (NCT04663347). Preliminary results presented at the 2021 ASH Annual Meeting suggest that the combination might be safe for the frontline treatment of DLBCL.

#### 7.1.3. Mosunetuzumab

Mosunetuzumab is a CD3xCD20 T-cell-dependent bispecific antibody (CD20-TDB), which is a full-length, humanized immunoglobulin G1 molecule with near-native antibody architecture constructed using “knobs-in-holes” technology (Figure 2b). Results of a phase I/II trial in 129 patients with R/R aggressive B-NHL, including 82 with de novo DLBCL, demonstrated an ORR of 34.9% with a median DoR and PFS of 7.6 and 1.4 months, respectively. CR rate was restricted to 19.4%; however, complete responses appeared to be rather durable as the median duration of CR was estimated to be 22.8 months, with 70.8% of patients who achieved CR remaining in remission up to 16 months off treatment. As described in Table 6, a step-up schedule was adopted to mitigate adverse events. Indeed, the most common grade 3 adverse event was neutropenia, whereas CRS of grade ≥ 3 emerged in 2 patients. Grade 3 neurologic adverse events occurred in 4.1% of patients; however, only two of them (1.0%) were considered treatment-related. Notably, two fatalities attributed to treatment have been noted, a case of hemophagocytic lymphohistiocytosis and a case of pneumonia [126]. Interestingly, among 19 patients who had previously undergone CAR T-cell therapy, ORR and CR rate were 36.8% and 26.3%, respectively, rendering this agent an attractive option for treatment after CAR T-cell failure; however, the duration of response could not be estimated due to the small sample size [131].

Recently preliminary results regarding subcutaneous administration of mosunetuzumab were presented. In a phase I/II study of 74 R/R B-NHL patients, including 31 DLBCL, it was shown that a step-up schedule of 5/45/45 mg during cycle 1 was safe with low CRS rates. Most importantly, ORR was 35.3%, which converges with the respective estimate for intravenous mosunetuzumab [132]. Mosunetuzumab is being evaluated in several trials in combination with gemcitabine and oxaliplatin (NCT04313608), atezolizumab (NCT02500407), or PolaV (NCT03671018). Interim results of the latter phase Ib/II demonstrated an ORR of 65% with 48.3%, with median DoR not reached at a median follow-up of 5.3 months. Most importantly, the combination demonstrated the same efficacy in a subgroup of 24 patients with prior CAR T-cell therapy failure.

In the frontline setting, mosunetuzumab is being evaluated as monotherapy in elderly or unfit previously untreated patients (NCT03677154); preliminary results, presented in ICML 2021, on 40 patients demonstrated an ORR of 67.7%, including 41.9% CR. Of 13 patients with CR, 4 durable responses were seen for ≥12 months from therapy initiation. Notably, no grade 3 CRS or neurological events were noted; therefore, monotherapy with this agent might be an appealing chemotherapy-free treatment option for elderly unfit patients [49]. Another study is evaluating mosunetuzumab in combination with CHOP or PolaV-CHP in newly diagnosed DLBCL patients (NCT03677141); preliminary results in 36 patients who received mosunetuzumab in combination with CHOP as first-line treatment demonstrated an ORR of 96.3% with 85.2% of patients achieving CR, grade 3 neutropenia was observed in 64% patients, whereas no grade 3 CRS or neurological events were noted [50]. These results are summarized in Table 2. Based on these results, mosunetuzumab might be an effective and safe therapeutic modality for treating DLBCL patients combined with conventional chemotherapy, such as CHOP.

#### 7.1.4. Odronextamab

Odronextamab (REGN1979) is a fully-humanized anti-CD20xanti-CD3 bispecific IgG4 antibody with a relatively long half-life (14 days) (Figure 2e). Interim results of a phase I trial on 127 R/R B-NHL, including 71 DLBCL patients, demonstrated an ORR of 60% (60% CR) in the 10 patients treated with a dose ≥ 80 mg who had not undergone CAR T-cell therapy. The median duration of response and PFS were 10.3 and 11.1 months, respectively. Notably, the 21 patients who had undergone CAR T therapy had substantially lower response rates (ORR: 33.3%, CR: 23.8%), whereas the responses were short-lived, with a median duration of 2.8 months. The most common adverse events included pyrexia, CRS, and chills. CRS grade ≥ 3 occurred in 7.1% of patients, whereas grade 3 neurologic events were noted in 2.3% of patients. To overcome the dose-limiting effect of CRS, premedication with dexamethasone has been suggested, as well as administration in a step-up dosing schedule as described in Table 7 [125]. Given the aforementioned results, a phase II trial is currently ongoing (NCT03888105), along with a phase I trial assessing the combination of assessing odronextamab in combination with cemiplimab, a novel PD-1 inhibitor (NCT02651662). Interestingly a recent study demonstrated that high levels of baseline tumor-infiltrating T cells might be associated with clinical response to odronextamab, whereas baseline CD20 expression did not correlate with efficacy [133].

#### 7.1.5. Plamotamab

Plamotamab (XmAb13676) is another CD3xCD20 BsAb, constructed through an XmAb protein engineering platform (Figure 2f). It is currently investigated in the setting of R/R NHL and CLL. Preliminary results of a Phase I trial (NCT02924402) on 27 DLBCL patients demonstrated an ORR of 40.7%, with CR noted in 7 patients. The most common grade 3 adverse events were neutropenia, thrombocytopenia, pyrexia, and hypokalemia; CRS occurred in 11%, with 1 Grade 4 and no grade 3 events [127].

### 7.2. Targeting CD19

#### 7.2.1. Blinatumomab

Blinatumomab is the first and the only currently approved BiTE therapy for hematological malignancies. It consists of an antigen-binding site-directed against CD19 on the surface of B lymphocytes and a second one against CD3 on the surface of T lymphocytes (Figure 2l). Blinatumomab has been approved by the FDA and EMA to treat R/R B-ALL. The success of blinatumomab in B-ALL triggered the evaluation of this agent in several phase I and II trials, as described in Table 7. Despite the satisfactory and durable response rates, blinatumomab has not been selected for further development for DLBCL, as its short half-life requires continuous infusion and, therefore, hospitalization for long periods. Moreover, blinatumomab therapy is accompanied by high rates of neurotoxicity and CRS. On this ground, therapeutic modalities with a more favorable safety profile and the option to be given in an outpatient clinic might be preferable for the treatment of DLBCL. Nonetheless, an ongoing phase II/III trial (NCT02910063) examines blinatumomab as second salvage therapy in patients with R/R NHL. Interim results demonstrated an ORR of 37% and a complete metabolic response rate of 22%. Grade 3 adverse events were reported in 59% of the patients, with 56% experiencing neurologic events of any grade [130]. There are currently three ongoing trials evaluating blinatumomab in combination with lenalidomide (NCT02568553) or pembrolizumab (NCT03340766) as well as post-allogeneic SCT maintenance (NCT03114865).

#### 7.2.2. Other CD19 Targeting BsAbs

Several other BsAbs are currently in the early stages of development. Among them, AMG 562, a CD3xCD19 BiTE with an extended half-life because of an added single-chain crystallizable fragment (scFc), is currently investigated in a phase I trial (NCT03571828) in R/R NHL patients [134]; A-319, another CD3xCD19 BsAb, is investigated in the same setting (NCT04056975) [135]. On the other hand, the development of two BsAbs with anti-CD19 activity, namely AFM11, a humanized tetravalent bispecific CD3xCD19 tandem diabody (TandAb) and duvortuxizumab (JNJ-64052781 or MGD011) a CD3xCD19 DART protein, has been halted due to unacceptable neurotoxicity [136,137].

### 7.3. Targeting CD47

As previously described, CD47 represents an appealing potential target for novel treatments in DLBCL; however, its abundancy in normal tissues prevents the antibodies from reaching their target. BsAbs have been developed to circumvent this issue with reduced affinity for CD47 and high affinity to a second tumor antigen. In this context, TG-1801 or NI-1701, a CD47xCD19 BsAb, has been developed and is currently undergoing a phase I trial (NCT03804996) [138]. Similarly, RTX-47 is a CD47xCD20 BsAb, consisting of a CD47 blocking scFv fused in tandem to a CD20 targeting scFv derived from rituximab [139], which could potentially be useful in the treatment of DLBCL; however, no human trials are yet available.

### 7.4. Bispecific ADCs

DT2219, OXS-1550, or DT2219ARL is a bispecific scFv recombinant fusion protein-drug conjugate composed of the variable regions of the heavy and light chains of anti-CD19 and anti-CD22 antibodies and a modified form of diphtheria toxin as its cytotoxic drug payload [140]. In a phase I/II trial, 18 patients with R/R NHL or ALL were evaluated. Three out of 13 evaluated patients had an objective response. Capillary leak syndrome, elevation in liver function test, and thrombocytopenia were among the most common grade ≥ 3 adverse events [141].

## 8. Conclusions

During the last five years, there has been tremendous progress in the treatment of DLBCL with the approval of several new agents and the development of much more that might get approval in the near future. These advancements might change the treatment landscape of DLBCL radically. Among the newly approved monoclonal antibodies, polatuzumab vedotin, highly effective and safe in combination with BR, may become the standard of care for the treatment of R/R DLBCL ineligible for ASCT. Moreover, PolaV might represent a potent bridging therapy to more radical approaches such as allo-HSCT or CAR T-cell therapy. Notably, the findings of the POLARIX trial in the frontline setting, demonstrating that PolaV in combination with R-CHP significantly prolongs PFS compared to conventional R-CHOP, might signify the revision of treatment guidelines even for the newly-diagnosed patients with DLBCL. Tafasitamab, in combination with lenalidomide, represents a potent chemotherapy-free approach for the treatment of R/R patients. Given that the survival curves present a plateau, a small portion of patients might get cured with this combination. Lastly, loncastuximab can be used as monotherapy for heavily pretreated DLBCL patients, even after CAR T cell therapy failure. Phase III studies are eagerly anticipated for tafasitamab and loncastuximab tesirine in the frontline setting.

Undoubtedly, the most exciting milestone in treating DLBCL so far has been the development of CAR T-cells. Based on the recent studies, CAR T-cells represent the most promising and innovative agents in the treatment of aggressive lymphomas; however, longer follow-up, a better understanding of prognostic factors as well as effective management of adverse events such as cytokine release syndrome and neurological complications, are required; moreover, factors pertaining to the high cost and logistics should be taken into consideration. Most importantly, CAR T-cells are not readily available as apheresis and construction of the product require several weeks. During this period, the patient might succumb to the disease, particularly in the absence of a potent bridging therapy.

Regarding agents under investigation, several CD3xCD20 bispecific antibodies are being evaluated in clinical studies pertaining to both the R/R and the frontline setting. As reports regarding their efficacy and safety are accumulating, we believe that several of these agents will get approved in the near future, further optimizing the treatment of DLBCL patients and also allowing more patients to have access to more radical treatment modalities, such as ASCT, allo-HSCT or CAR T-cell treatment. Notably, CD3xCD20 BsAbs have a more appealing dosing schedule and route of administration as their structure does not require continuous infusion, as opposed to blinatumomab. More striking, these agents remain effective after both ASCT and CAR-T cell therapy. Preliminary results have shown a plateau in survival curves; therefore, these agents might offer a cure in a proportion of R/R patients. Surprisingly, trials with CD3xCD20 BsAbs have demonstrated low rates of neurotoxicity in terms of immune effector cell-associated neurotoxicity syndrome (ICANS), vastly divergent from the high rates noted with blinatumomab and CD19 CAR T-cell therapies. A potential explanation might be that CD19 CAR T-cell and blinatumomab diminish CD19-expressing pericytes leading to endothelial activation and increased blood-brain barrier permeability [142]. In line with this hypothesis might be the observation of no ICANS in trials investigating non-CD19 CAR T [143,144]. Nevertheless, low levels of CRS, mediated by step-up dosing, and low ICANS rates are among the advantages of CD3xCD20 BsAbs over CAR T cells therapies.

In conclusion, the recent years have been marked by a vigorous development of novel treatment modalities for DLBCL and aggressive lymphomas in general. However, it should be noted that DLBCL represents a highly heterogeneous disease entity. Recent studies have unveiled distinct genetic subgroups within the DLBCL that could benefit from different therapeutic agents. On this ground, studies should focus on discovering novel biomarkers that might predict the response to each agent to permit a more personalized treatment approach, especially for high-risk or highly refractory patients. Treatment guidelines should also be updated to incorporate novel agents and specify the optimal sequence in which these agents should be used.

## Figures and Tables

**Figure 1 cancers-14-01917-f001:**
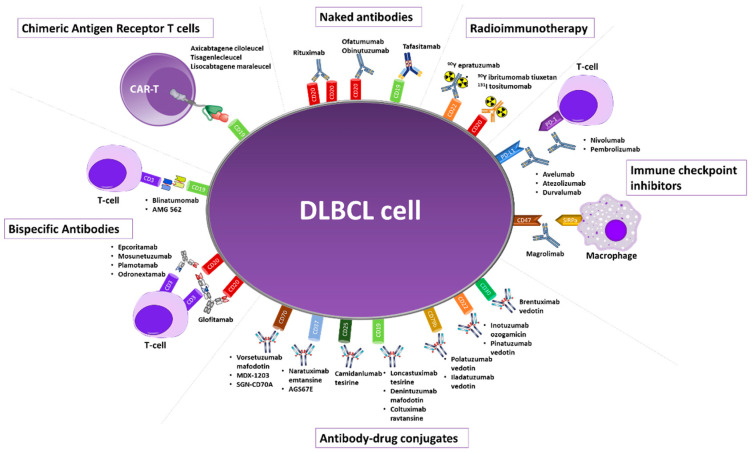
Overview of the monoclonal antibodies investigated for the treatment of diffuse large B-cell lymphoma.

**Figure 2 cancers-14-01917-f002:**
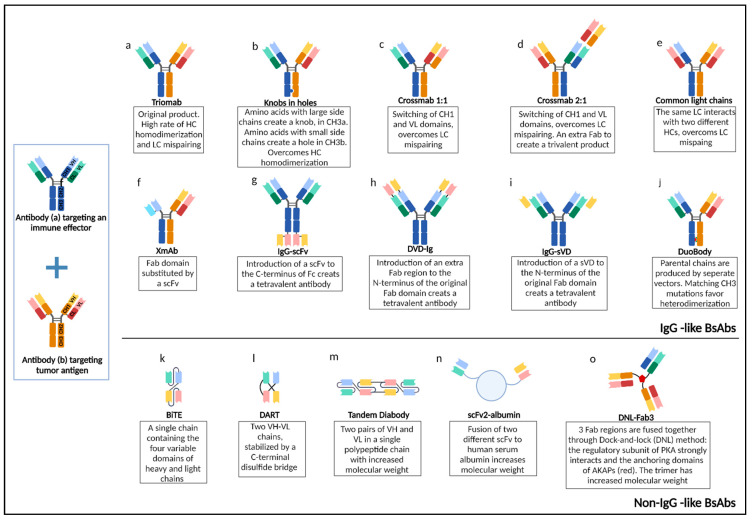
Structure of bispecific antibodies (BsAbs). Two pre-established hybridomas expressing antibodies (**a**,**b**) are fused to generate a quadroma that expresses both parental heavy chains (HCs) and light chains (LCs). The upper panel (**a**–**j**) summarizes the structure of IgG-like bispecific antibodies. Methods to circumvent HC homodimerization and LC mispairing are highlighted. The lower panel (**k**–**o**) depicts the structure of non-IgG-like bispecific antibodies, along with methods for the production of molecules with increased half-life. HC: heavy chain; LC: light chain CH1-3: Constant domains of heavy chain; VH: variable domain of heavy chain; CL: constant domain of light chain; VL: variable domain of light chain; Fc: fragment crystallizable region; Fab: fragment antigen-binding; ScFv: single-chain variable fragment; sVD: single variable domain; DVD-Ig: dual-variable domain immunoglobulin; BiTE: bi-specific T-cell engager; DART: dual-affinity re-targeting antibody; DNL: dock and lock; PKA: cAMP-dependent protein kinase; AKAP: A-kinase anchor proteins (Created with BioRender.com).

**Table 1 cancers-14-01917-t001:** Overview of approved and investigational monoclonal antibodies in DLBCL treatment.

Name of Antibody	Target	Approved for Newly Diagnosed DLBCL	Approved for R/R DLBCL	Commercially Available (Off-Label for DLBCL)	Investigational
Naked Antibodies					
Rituximab	CD20	√			
Ofatumumab				√	
Obinutuzumab				√	
Enhanced antibodies					
Tafasitamab	CD19		√ ^†^		
Radioimmunotherapy					
^90^Y epratuzumab	CD22				√
^90^Y ibritumomab tiuxetan	CD20			√	
^131^I tositumomab				√	
Immune checkpoint inhibitors					
Nivolumab	PD-1			√	
Pembrolizumab			√ *^,†^		
Avelumab	PD-L1			√	
Atezolizumab				√	
Durvalumab				√	
Magrolimab	CD47				√
Antibody-drug conjugates					
Brentuximab vedotin	CD30			√	
Inotuzumab ozogamicin	CD22			√	
Pinatuzumab vedotin					√
Polatuzumab vedotin	CD79b		√		
Iladatuzumab vedotin					√
Loncastuximab tesirine	CD19		√ ^†^		
Denintuzumab mafodotin					√
Coltuximab ravtansine					√
Camidanlumab tesirine	CD25				√
Naratuximab emtansine	CD37				√
Vorsetuzumab mafodotin	CD70				√
Zilovertamab vedotin	ROR1				√
Bispecific antibodies					
Glofitamab	CD3xCD20				√
Epcoritamab					√
Mosunetuzumab					√
Plamotamab					√
Odronextamab					√
Blinatumomab	CD3xCD19			√	
Chimeric antigen receptor T-cells					
Axicabtagene ciloleucel	CD19		√		
Tisagenlecleucel			√		
Lisocabtagene maraleucel			√ ^†^		

^†^ Approved only by the FDA; * Approved only for Primary Mediastinal Large B-cell, R/R: relapsed/refractory.

**Table 2 cancers-14-01917-t002:** Clinical trials evaluating the efficacy of monoclonal antibodies in patients with newly diagnosed DLBCL.

Reference	Study ID	Study Phase	Study Population	Treatment Protocol	Status	ORR (CR), %	PFS	OS
Tafasitamab								
Belada (2020) [40]	First-MIND NCT04134936	Ib	66 (IPI 2–5, ECOG PS: 0–2)	Tafasitamab (12 mg/kg d1, 8, 15) + R-CHOP (C1–6)	Active, not recruiting	76 (NR)	NR	NR
				Tafasitamab + R^2^-CHOP (C1–6)		82 (NR)		
	Front-MIND NCT04824092	III	IPI 3–5 (>60 y) or aaIPI 2–3 (≤60 y) ECOG PS: 0–2	Tafasitamab (12 mg/kg d1, 8, 15) + R^2^-CHOP (C1–6)	Recruiting			
				R-CHOP (C1–6)				
Pembrolizumab								
Smith (2020) [41]	NCT02541565	I	24 (ECOG PS 0–1)	Pembrolizumab (200 mg q3w) + R-CHOP (C1–6)	Completed	90 (77)	83% (2 y)	84% (2 y)
Atezolizumab								
Younes (2019) [42]	NCT02596971	I/II	42 (ECOG PS 0–2)	Atezolizumab (1200 mg d1 C2–8) + R-CHOP Consolidation: atezolizumab 1200 mg q3w C9–25	Completed	88 (79)	75% (2 y)	86% (2 y)
Durvalumab								
Nowakowski (2021)	NCT03003520	II	46 (IPI ≥ 3, NCCN-IPI ≥ 4, ECOG PS 0–2)	Durvalumab (1125 mg d1 C1–8) + R-CHOP Consolidation: durvalumab 1500 mg q4w × 12 cycles	Active, not recruiting	97 (68)	68% (1 y)	NR
				Durvalumab + R^2^-CHOP Consolidation: durvalumab q4w × 12 cycles		100 (67)	67% (1 y)	NR
Avelumab								
Hawkes (2020) [43]	NCT03244176	ΙΙ	28 (Stage II-IV, ECOG: 0–2)	Avelumab (10 mg/kg) + rituximab q2w for two cycles followed by R-CHOP-21 × 6 cycles. Consolidation: avelumab q2w × 6 cycles	Recruiting	89 (89)	76% (1 y)	89% (1 y)
Brentuximab vedotin								
Budde (2016) [44]	NCT01925612	II	51 (IPI 3–5 or aaIPI 2–3	BV (1.2 or 1.8 mg/kg) + R-CHOP q3w	Terminated (portfolio prioritization)	NR (69)	CD30+: 79% CD30-: 58% (1.5 y)	CD30+: 92% CD30-: 71% (1.5 y)
			11 (IPI 3–5 or aaIPI 2–3)	BV 1.8 mg/kg + R-CHP q3w		91 (82)	NR	NR
Svoboda (2020) [45]	NCT01994850	II	29 (5 DLBCL, 22 PMLBL, 2 GZL)	BV 1.8 mg/kg + R-CHP q3w	Completed	100 (86)	Not reached 85% (2 y)	Not reached 100% (2 y)
Reagan (2020) [46]	NCT02734771	II	22 (age ≥ 75 years)	BV 1.8 mg/kg + R-miniCHP q3w	Active, not recruiting	86 (67)	Not reached 61% (2 y)	Not reached 74% (2 y)
Polatuzumab vedotin								
Tilly (2019) [47]	NCT01992653	Ib/II	66 (ECOG PS 0–2)	PolaV 1.8 mg/kg + R-CHP or G-CHP	Completed	89 (77)	83% (2 y)	94% (1 y)
Tilly (2021) [48]	POLARIX NCT01992653	III	879 (IPI 2–5, ECOG PS 0–2)	PolaV 1.8 mg/kg + R-CHP	Completed	86 (78)	77% (2 y) *	89% (2 y)
				R-CHOP		84 (74)	70 (2 y) *	89% (2 y)
	NCT04231877	I	DLBCL, High-grade BCL	PolaV + R-da-EPOCH q3w	Active, not recruiting			
	NCT04479267	II	Double or triple-hit high-grade BCL	PolaV 1.8 mg/kg + R-CHP	Recruiting			
Glofitamab								
	NCT03467373	I	ECOG PS: 0–3	Glofitamab + R-CHOP	Recruiting			
Epcoritamab								
	NCT04663347	I/II	DLBCL	Epcoritamab + R-CHOP	Recruiting			
Mosunetuzumab								
Olszewski (2021) [49]	NCT03677141	I/II	40 DLBCL (elderly or unfit patients)	Monotherapy (d1, d8, and d15 of C1, then q3w for eight cycles)	Recruiting	68 (42)	NR	NR
Phillips (2020) [50]	NCT03677141	I/II	43 DLBCL	Mosunetuzumab + CHOP	Recruiting	96 (85)	NR	NR

* Statistically significant difference in PFS (p: 0.02). PFS: progression-free survival; OS: overall survival; NHL: non-Hodgkin lymphoma; DLBCL: diffuse large B-cell lymphoma; GZL: gray-zone lymphoma; PMLBCL: primary mediastinal large B-cell lymphoma; PTLD: post-transplant lymphoproliferative disorder; FL: follicular lymphoma; PBL: plasmablastic lymphoma; TCHRLBL: T-cell histiocyte rich large B-cell lymphoma; ASCT: Autologous hematopoietic stem cell transplantation; BV: brentuximab vedotin; PolaV: polatuzumab vedotin; NR: not-reported; IPI: International Prognostic Index; aaIPI: age-adjusted IPI; NCCN-IPI: National Comprehensive Cancer Network IPI; R-CHOP: rituximab, cyclophosphamide, doxorubicin, vincristine, prednisone; R^2^-CHOP: rituximab, lenalidomide, cyclophosphamide, doxorubicin, vincristine, prednisone; R-CHP: rituximab, cyclophosphamide, doxorubicin, prednisone; R-daEPOCH: rituximab, dose-adjusted etoposide, doxorubicin, vincristine, prednisone; qXw: every X weeks.

**Table 3 cancers-14-01917-t003:** Structure mechanism of action and key toxicities of commercially available and investigational antibody-drug conjugates.

ADC	Antibody	Linker	Payload Release ^†^	Payload	Target	DAR	Mode of Action ^§^	Payload Specific Toxicities
Brentuximab vedotin (BV)	Chimeric IgG1	Cleavable, MC-VC-PAB	I	MMAE	CD30	4	A	Peripheral neuropathy, neutropenia
Inotuzumab ozogamicin (InO)	Humanized IgG4	Cleavable, AcBut	II	N-acetyl gamma calicheamicin	CD22	6	B	Hepatotoxicity, veno-occlusive liver disease, thrombocytopenia
Polatuzumab vedotin (PolaV)	Humanized IgG1	Cleavable MC-VC-PAB	I	MMAE	CD79b	3.5	A	Peripheral neuropathy, neutropenia
Loncastuximab tesirine	Humanized IgG1	Cleavable, MC-PEG-VA	I	SG3199, PBD	CD19	2.3	C	Photosensitivity, increased GGT, myelosuppression, effusions, edema
Denintuzumab mafodotin (SGN-CD19A)	Humanized IgG1	Non-cleavable maleimidocaproyl linker	III	MMAF	CD19	NA	A	Ocular toxicity, Keratopathy
Coltuximab ravtansine (SAR3419)	Humanized IgG1	Cleavable, SPDB	IV	Maytansinoid DM4	CD19	3.5	A	Ocular toxicity
Pinatuzumab vedotin (DCDT2980S)	Humanized IgG1	Cleavable, MC-VC-PAB	I	MMAE	CD22	3.6	A	Peripheral neuropathy, neutropenia
Camidanlumab tesirine (ADCT-301)	Humanized IgG1	Cleavable MC-PEG-VA	I	SG3199, PBD	CD25	2.3	C	Photosensitivity, increased GGT, myelosuppression, effusions, edema
Naratuximab emtansine (IMGN529)	Humanized IgG1	Non-cleavableSMCC	III	Maytansinoid DM1	CD37	3–4	A	Thrombocytopenia, elevated liver transaminases, peripheral neuropathy
AGS67E	Humanized IgG2	Cleavable, MC-VC-PAB	I	MMAE	CD37	3.7	A	Peripheral neuropathy, neutropenia
MDX-1203	Humanized IgG1	Cleavable, MC-VC	I	MED-A Duocarmycin carbamate prodrug	CD70	1.25	C	Hypersensitivity, Effusions, edema
Vorsetuzumab mafodotin (SGN-75)	Humanized IgG1	Non-cleavable Maleimidocaproyl linker	III	MMAF	CD70	NA	A	Thrombocytopenia, Ocular toxicity
SGN-CD70A	Humanized IgG1	Cleavable, MC-VA	I	SGD-1882 PBD	CD70	2	C	Thrombocytopenia
Iladatuzumab vedotin (DCDS0780A)	Humanized IgG1	Cleavable, MC-VC-PAB	I	MMAE	CD79b	2	A	Ocular toxicity, neutropenia
Zilovertamab Vedotin (MK-2140)	Humanized IgG1	Cleavable, MC-VC-PAB	I	MMAE	ROR1	4	A	Peripheral neuropathy, neutropenia

^†^ Mechanisms of Payload release: I: Linker cleaved by lysosomal cathepsin B; II: Linker hydrolyzed in acidic endosome environment; III: Payload released following complete antibody degradation; IV: Linker cleaved by glutathione, ^§^ Mechanisms of ADC action: A: Tubulin polymerization inhibition, microtubule structures destabilization; B: DNA double-strand cleavage; C: Covalently binding to the minor groove of the double-stranded DNA. DNA adducts inhibit DNA replication causing cell-cycle arrest and apoptosis. ADC: antibody-drug conjugate; DAR: drug to antibody ratio; MC-VC-PAB: maleimidocaproyl-valine-citrulline-p -aminobenzoyloxycarbonyl; AcBut: 4-(4-acetylphenoxy) butanoic acid; MC-PEG-VA: maleimidocaproyl-polyethylene glycol- valine-alanine; SPDB: N-Succinimidyl 4-(2-pyridyldithio) butanoate; MC-VA: maleimidocaproyl- valine-alanine; MC-VC: Maleimidocaproyl-valine-citrulline; SMCC: succinimidyl-4-(N-maleimidomethyl)-cyclohexane-1-carboxylate; MMAE: Monomethyl auristatin E; PBD: Pyrrolobenzodiazepine dimer; MMAF: Monomethyl auristatin F.

**Table 4 cancers-14-01917-t004:** Clinical trials evaluating the efficacy of commercially available antibody-drug conjugates in R/R DLBCL.

Reference	Study Phase	Patients	Age, Median (Range)	Prior Treatment Lines Median (Range)	Treatment Protocol	ORR (CR), %	mDOR (Months)	Median PFS (Months)
Brentuximab vedotin								
Jacobsen (2015) [71]	II	49 (CD30+, ECOG PS: 0–2)	62 (17–85)	3 (1–6)	BV 1.8 mg/kg q3w	44 (17)	5.6	4.0
		16 (CD30+, ECOG PS: 0–2)	62 (22–78)	3 (1–5)	BV 1.8 mg/kg q3w + rituximab q3w	46 (15)	NR	NR
Bartlett (2017) [72]	II	52 CD30u, ECOG PS: 0–2)	65 (21–91)	2 (1–4)	BV 1.8 mg/kg q3w	31 (12)	4.7	1.4
Kim (2019) [73]	II	12 (CD30 > 30%, ECOG PS: 0–2)	56 (27–73)	3 (2–10)	BV 1.8 mg/kg q3w	50 (17)	6.0	1.9
Zinzani (2017) [74]	II	15 PMLBCL (CD30+, ECOG PS: 0–1)	29 (20–73)	3 (1–4)	BV 1.8 mg/kg q3w	13 (0)	<3	NR
Ward (2021) [75]	I	37 (ECOG PS: 0–2)	65 (51–79)	3 (1–6)	BV 1.8 mg/kg q3w + lenalidomide 20 mg per day	57 (35)	13.1	10.2
Zinzani (2019) [76]	I/II	30 PMLBCL (CD30+, ECOG PS: 0–1)	36 (19–83)	2 (2–5)	BV 1.8 mg/kg q3w + nivolumab q3w	73 (37)	31.6	26.0
Inotuzumab ozogamicin								
Advani (2010) [78]	I	35 (CD22+, ECOG PS: 0–2)	60 (26–82)	≥4: 61%	InO 1.8 mg/m^2^ q3w	15 (8)	NR	1.6
Fayad (2013) [79]	I/II	47 (CD22+, ECOG PS: 0–2)	72 (32–85)	≥3: 9%	InO 1.8 mg/m^2^ + rituximab q4w	74 (50)	17.7	17.1
Wagner-Johnston (2015) [80]	II	63 (CD22+, ECOG PS: 0–2, Prior Tx lines ≤ 2)	60 (19–75)	≥2: 52	InO 1.8 mg/m^2^ + rituximab 375 mg/m^2^ q3w ± ASCT	30 (14)	NR	3.0
Dang (2018) [81]	III	338 (CD22+, ECOG PS: 0–3, ASCT ineligible)	70 (18–92)	2 (1–3)	InO 1.8 mg/m^2^ + rituximab q4w	41 (16)	11.6	3.7
					BR or R-G	44 (16)	6.9	3.5
Ogura (2016) [82]	I	21 (CD22+, ECOG PS: 0–2)	63 (42–81)	2 (1–6)	InO 0.8 mg/m^2^ + R-CVP q3w	57 (10)	11	4
Sangha (2017) [83]	I	21 (CD22+, ECOG PS: 0–2)	65 (25–81)	2 (1–6)	InO 0.8 mg/m^2^ + R-GDP q3w	33 (19)	9.3	6.1
Polatuzumab vedotin								
Palanca-Wessels (2015) [84]	I/II	25 (ECOG PS: 0–2)	67 (20–81)	≥3: 88%	PolaV 2.4 mg/kg q3w	56 (16)	5.2	5.0
					PolaV 2.4 mg/kg + rituximab q3w	78 (22)	12.3	12.5
Morschhauser (2019) [85]	Ib/II	39 (ECOG PS: 0–2)	68 (55–77)	3 (2–6)	PolaV 2.4 mg/kg + rituximab q3w	54 (21)	13.4	5.6
Phillips (2016) [86]	Ib/II	38 (ECOG PS: 0–2)	71 (27–84)	2 (1–7)	PolaV 2.4 mg/kg + obinutuzumab q3w	52 (29)	NR	NR
Sehn (2020) [87]	Ib/II	40 (ECOG PS: 0–2)	67 (33–86)	2 (1–7)	PolaV 1.8 mg/kg + BR q3w	45 (40)	12.5	9.5
Sehn (2021) [88]	Ib/II	106 (ECOG PS: 0–2)	70 (24–94)	2 (1–7)	PolaV 1.8 mg/kg + BR q3w	42 (39)	9.5	6.6
Loncastuximab tesirine								
Hamadani (2020) [89]	I	139 DLBCL	63 (20–86)	3 (1–10)	0.15 mg/kg q3w	42 (23)	4.5	2.8
Caimi (2021) [90]	II	145 DLBCL	66 (56–71)	3 (2–4)	0.15 mg/kg q3w for 2 cycles and then 0.075 mg/kg for one year	48 (24)	12.6	4.9

NHL: non-Hodgkin lymphoma; DLBCL: diffuse large B-cell lymphoma; GZL: gray-zone lymphoma; PMLBCL: primary mediastinal large B-cell lymphoma; PTLD: post-transplant lymphoproliferative disorder; FL: follicular lymphoma; PBL: plasmablastic lymphoma; ASCT: Autologous hematopoietic stem cell transplantation; BV: brentuximab vedotin; InO: Inotuzumab ozogamicin; R/R: relapsed/refractory; ICI: immunochemotherapy; CD30u: CD30 undetectable; NR: not-reported; IPI: International Prognostic Index; aaIPI: age-adjusted IPI; R-CHP: rituximab, cyclophosphamide, doxorubicin, prednisone; R-CVP: rituximab, cyclophosphamide, vincristine, prednisone; R-GDP: rituximab, gemcitabine, dexamethasone, cisplatin; BR: rituximab, bendamustine; R-G: rituximab, gemcitabine; mDOR: median duration of response qXw: every X weeks.

**Table 5 cancers-14-01917-t005:** Clinical trial and real-life data evaluating the efficacy of polatuzumab vedotin in combination with bendamustine and rituximab in patients with relapsed refractory diffuse large B-cell lymphoma.

	Dimou et al. [99]	Segman et al. [100]	Smith et al. [101]	Northend et al. [102]	Liebers et al. [97]	Vodicka et al. [103]	Sehn et al. [88]
**Patients (n)**	49	32	69	133	54	21	106
**Age, median (range)**	63 (67–85)	66 (37–77)	62 (17–88)	72 (18–88)	74 (37–87)	67 (35–85)	70 (24–94)
**Gender (male %)**	53	60	62	65.4	68.5	NA	49
**Median prior treatment lines (range)**	2 (1–9)	3 (2–3)	3 (1–9)	≥2: 64.7	3 (2–8)	3 (2–7)	2 (1–7)
**Refractory to 1st treatment line (%)**	67	53	NA	NA	NA	NA	69
**Refractory to last treatment line (%)**	38 (78)	23 (72)	NA	68.4	87	NA	76
**Performance status, %**	0–1: 61 2: 20 3–4: 20	≥2: 53	>1: 33	≥2: 30	NA	≥2: 76	0: 28 1: 59 2: 13
**Ann-Arbor stage III/IV (%)**	57	88	NA	NA	NA	48	79
**Prior ASCT, %**	16	31	16	NA	9	29	18
**Prior CAR-T cell therapy, %**	0	3	26	NA	9	NA	1
**ORR, %**	43	63	50	57	48	33	42
**CR, %**	25	38	24	33	15	24	39
**Median PFS (95% CI) (months)**	4 (2.3–5.8)	5.6 (2.97–7.97)	2 (NA)	4.8 (3.7–9.3)	1 y: 8.0% (1.7–38.3)	8.7 (NA)	6.6 (5.1–9.2)
**Median OS (95% CI) (months)**	8.5 (3.1–13.8)	8.3 (5.4–14.8)	5.3 (NA)	8.2 (5.9–14.3)	1 y: 12.6 (4.1–38.9)	3.8 (NA)	12.5 (8.3–23.1)
**Median DoR (95% CI) (months)**	8.7 (4–16)	12.6 (NA)	5 (NA)	NA	NA	NA	9.5 (7.9–23.1)

ASCT: Autologous hematopoietic stem cell transplantation; ORR: objective response rate; CR: complete remission; PFS: Progression-free survival; OS: overall survival; DOR: duration of response; 95%CI: 95% confidence intervals; NA: Not available.

**Table 6 cancers-14-01917-t006:** Comparison of structure, route of administration, and treatment schedule of CD3xCD20 bispecific antibodies.

	Glofitamab Hutchings et al. [123]	Epcoritamab Hutchings et al. [124]	Odronextamab Bannerji et al. [125]	Mosunetuzumab Budde et al. [126]
**Structure**	IgG-like Humanized IgG1 CrossMab (2:1)	IgG-like Humanized IgG1 DuoBody (1:1)	IgG-like Humanized IgG4 Common light chains (1:1)	IgG-like Humanized IgG1 Knobs-in-holes (1:1)
**Route of administration**	IV	SC	IV	IV or SC
**Plasma half-life**	6–11 days	8.67 days	14 days	6–11 days
**Recommended phase 2 dose (RP2D)**	2.5/10/30 mg	48 mg	160/320 mg	1/2/60/60/30 mg (IV) 5/45/45 mg (SC)
**Treatment Schedule**	C1d1: Obinutuzumab 1000 mg C1d8: Glofitamab 2.5 mg C1d15: Glofitamab 10 mg C2d1-C12d1: Glofitamab 30 mg (Cycles of 21 days)	C1-2: qw C3-6: q2w C7-: q4w (Cycles of 28 days)	160 mg qw for 12 weeks 320 mg q2w onwards Week 1: Initial dose split in two days Week 2: Intermediate dose split in two days	C1d1: 1 mg (IV) 5 mg (SC) C1d8: 2 mg (IV) 45 mg (SC) C1d15: 60 mg (IV) 45 mg (SC) C2d1: 60 mg (IV) 45 mg (SC) C3d1-: 30 mg (IV) 45 mg (SC) (Cycles of 21 days)
**Duration of treatment**	Maximum of 12 cycles unless disease progression or unacceptable toxicity	Until disease progression or unacceptable toxicity	Until disease progression or unacceptable toxicity In patients with durable CR: q4w	Maximum of 17 cycles unless disease progression or unacceptable toxicity Discontinuation after 8 cycles in case of CR

**Table 7 cancers-14-01917-t007:** Efficacy and safety of bispecific antibodies evaluated in clinical trials for the treatment of R/R DLBCL.

Bispecific Antibody	Study PHASE	Patients	Age, Median (Range)	Prior Treatment Lines, Median (Range)	Treatment Protocol	Efficacy	Adverse Events (Grade ≥ 3)
CD3xCD20							
Glofitamab [123]	I	258 (98 DLBCL)	64 (22–86)	3 (1–12)	Fixed dosing: 0.6–25 mg q2w or q3w Step-up dosing: 2.5/10/16 mg or 2.5/10/30 mg q3w	ORR: 54% CR: 39% mDOR: 29.4 months mPFS: 2.9 months	Neutropenia (25%) Thrombocytopenia (8%) Anemia (8%) CRS (5%)
Epcoritamab [124]	I/II	68 (46 DLBCL)	68 (55–74)	3 (2–4)	0.76–48 mg q1w: cycle 1–2; q2w: cycle 3–6; q4w thereafter	ORR: 68%; CR: 46%	Anemia (13%) Fatigue (6%) Hypotension (6%) Neurotoxicity (3%)
Mosunetuzumab [126]	I	129 (82 DLBCL)	63 (19–91)	3 (1–14)	1/2/60/60/30 mg q3w	ORR 35%; CR 19% mDOR: 7.6 months mPFS: 1.4 months	Neutropenia (25%) Hypophosphatemia (15%) Anemia (9%) CRS (1%) Neurotoxicity (4%)
Odronextamab [125]	I	127 (71 DLBCL)	NR	3 (1–11)	0.03–320 mg q1w for 12w followed by q2w	No prior CAR-T: ORR: 60% CR: 60% mDOR: 10.3 months mPFS: 11.2 months Prior CAR-T: ORR: 33% CR: 23.8% mDOR: 2.8 months mPFS: 2.5 months	CRS (7%) Neurotoxicity (2%)
Plamotamab [127]	I	47 (27 DLBCL)	62 (32–89)	4 (1–10)	20–125 μg/kg q1w	ORR 41% CR: 26%	Neutropenia (14%) Thrombocytopenia (8%) Hypokalemia (6%) CRS (3%)
CD3xCD19							
Blinatumomab [128]	I	76 (14 DLBCL)	65 (20–80)	3 (1–10)	60 μg/m^2^/d Continuous IV 4–8 weeks followed by a 4-week consolidation	ORR: 55% CR: 36% mDOR: 13.5 months	Lymphopenia (79%) Neurologic (22%) Increase CRP (20%) Leukopenia (20%) Neutropenia (17%) Thrombocytopenia (12%) Hyperglycemia (12%)
Blinatumomab [129]	II	25 DLBCL	66 (34–85)	3 (1–7)	112 μg/d Continuous IV infusion for 8 weeks followed by 4-week consolidation	ORR 43% CR 19% mDoR 11.6 months mPFS 3.7 months	Thrombocytopenia (17%) Leukopenia: 17% Neurotoxicity (22%)
Blinatumomab [130]	II/III	41 (34 DLBCL)	59 (19–75)	NR	9 μg/d for 7 d 28 μg/d for 7 d, 112 μg/d for 42 d followed by an optional 4-week cycle.	ORR: 37%; CR: 22% mPFS: 2.5 months mOS: Not reached	Neutropenia (10%) Neurotoxicity (24%)

NHL: non-Hodgkin lymphoma; aNHL: aggressive NHL; DLBCL: diffuse large B-cell lymphoma; IV: intravenous; SC: subcutaneous; ORR: objective response rate; CR: complete remission; PR: partial response; mDOR: median duration of response; mPFS: median progression-free survival; CAR-T: chimeric antigen receptor T-cells: CRS: cytokine release syndrome; LFT: liver function tests; qXw: every X weeks; dX: day X.

**Table 8 cancers-14-01917-t008:** Characteristics of studies evaluating CD3xCD20 bispecific antibodies in the treatment of R/R DLBCL.

	Glofitamab Hutchings et al. [123]	Epcoritamab Hutchings et al. [124]		Odronextamab Bannerji et al. [125]		Mosunetuzumab Budde et al. [126]
Study Phase	Ι	I/II		Ι		I
**Patients (n)**	258 98 DLBCL 31 tFL, 26 MCL, 11 RT 75 iNHL	68 46 DLBCL 12 FL 4 MCL		127 71 DLBCL 37 FL 11 MCL 6 MZL 2 Β-NHL		129 82 DLBCL 26 tFL 13 MCL 5 RT 1 FL gr 3B
**Age, median (range)**	64 (22–86)	68 (55–74)		ΝR		63 (19–91)
**Eligibility criteria**	R/R B-NHL, ECOG PS ≤ 1	R/R B-NHL, ECOG PS ≤ 1 (after at least one Tx with anti-CD20 mAb)		R/R B-NHL, ECOG PS ≤ 1 (after at least one Tx with anti-CD20 mAb)		R/R B-NHL, ECOG PS ≤ 1
**Prior treatment lines, median (range)**	3 (1–12)	3 (2–4)		3 (1–11)		3 (1–14)
**Prior CAR-T cells (#)**	3 (1.8%)	5 DLBCL		29 (23%) (25 DLBCL)		15 (11.6%)
**Median follow-up (months)**	13.5	10.2		3.9		11.9
**Premedication (CRS prophylaxis)**	1000 mg obinutuzumab on day -7	Corticosteroids		Dexamethasone		Corticosteroids
**Dose**	2.5/10/16 mg 2.5/10/30 mg	≥12 mg (*n* = 18)	≥48 mg (*n* = 7)	Dose ≥ 80 mg		1/2/60/60/30 mg
				Prior CAR-T (*n* = 21)	No prior CAR-T) (*n* = 10)	
**ORR (%)**	54	68	91	33	60	35
**CR (%)**	39	45	55	24	60	19
**Median DOR (months)**	29.4	ΝR	ΝR	2.8 DoCR: 4.4	10.3 DoCR: 9.5	7.6 DoCR: 22.8
**Median PFS (months)**	2.9	9.1	Not reached	2.5	11.1	1.4
**CRS (%) (grade 3 + 4)**	5.1%	0		7.1% (6.3 + 0.8)		1%
**Neurotoxicity (grade 3 + 4)**	0	3%		2.3% (2.3 + 0)		4.1%
**Treatment discontinuation**	2.9%	0		5.5%		3.6%

B-NHL: B non-Hodgkin lymphoma; DLBCL: diffuse large B-cell lymphoma; tFL: transformed follicular lymphoma; MCL: mantle-cell lymphoma; iNHL: indolent NHL; RT: Richter transformation; MZL: marginal-zone lymphoma; BCL: B-cell lymphoma; Tx: treatment; ECOG PS: Eastern Cooperative Oncology Group performance status; IC: immunochemotherapy; CRS: cytokine release syndrome; ORR: objective response rate; CR: complete remission; PFS: progression-free survival: OS: overall survival: DOR: duration of response; DoCR: duration of complete remission; NR: not reported.

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
