# Peer review of "Monoclonal Antibodies in the Treatment of Diffuse Large B-Cell Lymphoma: Moving beyond Rituximab"

_cancers, 2022, doi:10.3390/cancers14081917_

Round 1

Reviewer 1 Report

Overall, this is a very well summarized and coherently written narrative review article. Also, as the title suggests, it is a review article that systematically describes and analyzes the summary of the latest monoclonal antibodies research results and the results of key clinical trials.

However, the arrangement and description of CAR T cell therapy could not have been included in this review article, considering the title of this review article and the length of the review article (this review article is too much longer than a normal review article). Personally, I think it would be better to deal with it in a separate review article on CAR T.

If the author wishes to include CAR T therapy in this review article, it is recommended that the title be changed from monoclonal antibodies to another to avoid confusion among readers.

Minor points;

  1. reference number

There are too many references compared to a normal narrative review article. In a typical narrative review article, around 100 references are recommended. If possible, you can reduce the number by citing references mainly for core and essential research rather than all references, and I think that other references can be found by the readers themselves through the cited references.

  1. Table 4, 5, 6

Tables 4, 5, and 6 seem to be fine without them. Studies that have already been described in detail in the text and have not entered the next stage of clinical research are considered to be sufficient as the text description.

  1. 'he' in the main text Line 663 should probably be replaced with 'The'.

  1. At this point, reference 53 has been published, so please replace the reference with the following:

N Engl J Med. 2022 Jan 27;386(4):351-363. doi: 10.1056/NEJMoa2115304.

  1. There are disproportionately too many explanations for polar-V compared to other monoclonal Abs. In order to avoid unnecessary misunderstandings, I recommend balancing the content of Pola-V by summarizing a little more.

Author Response

Response to Reviewer 1 Comments

Overall, this is a very well summarized and coherently written narrative review article. Also, as the title suggests, it is a review article that systematically describes and analyzes the summary of the latest monoclonal antibodies research results and the results of key clinical trials

We would like to thank the Reviewer for the encouraging comments that have helped to strengthen our manuscript

However, the arrangement and description of CAR T cell therapy could not have been included in this review article, considering the title of this review article and the length of the review article (this review article is too much longer than a normal review article). Personally, I think it would be better to deal with it in a separate review article on CAR T.

If the author wishes to include CAR T therapy in this review article, it is recommended that the title be changed from monoclonal antibodies to another to avoid confusion among readers.

We agree with the Reviewer that CAR T cell therapy, although exciting, it is beyond the scope of this review. In the revised manuscript we opted to focus on the development of monoclonal antibodies and eliminated the discussion on CAR T cells which could be discussed in a separate review.

Minor points;

Reference number

There are too many references compared to a normal narrative review article. In a typical narrative review article, around 100 references are recommended. If possible, you can reduce the number by citing references mainly for core and essential research rather than all references, and I think that other references can be found by the readers themselves through the cited references

Following the Reviewer’s suggestion, we reduced the number of references retaining only references pertaining to key studies.

Table 4, 5, 6

Tables 4, 5, and 6 seem to be fine without them. Studies that have already been described in detail in the text and have not entered the next stage of clinical research are considered to be sufficient as the text description.

We eliminated Table 6 as we agree that it was redundant. However, we believe that Table 4 and 5 should remain in the manuscript. Table 4 includes key finding on approved agents as polatuzumab vedotin and loncastuximab vedotin that are not fully-described within the manuscript, whereas Table 5 provides the real-world experience with PolaV. We believe that real-world data are important when evaluating the efficacy and safety of any given agent. Therefore a Table comparing real-world  data with that of the respective trial might be valuable for the reader.

'he' in the main text Line 663 should probably be replaced with 'The'.

This erratum has been corrected in the revised manuscript

At this point, reference 53 has been published, so please replace the reference with the following: N Engl J Med. 2022 Jan 27;386(4):351-363. doi: 10.1056/NEJMoa2115304.

The respective reference has been updated

There are disproportionately too many explanations for polar-V compared to other monoclonal Abs. In order to avoid unnecessary misunderstandings, I recommend balancing the content of Pola-V by summarizing a little more.

PolaV is the first of the novel agents that has been approved for the treatment of DLBCL and the only one that has undergone randomized trial both in the R/R and the frontline setting. Moreover, there has been substantial real-world experience with this agent. Therefore the available data that should be presented in this review exceed by far those available for loncastuximab tesirine and tafasitamab. Nonetheless, following the reviewer’s suggestion we provide a more concise discussion pertaining to PolaV as to provide a more balanced manuscript.

Reviewer 2 Report

Papageorgiou et al described novel anti-CD20 antibodies and new immunotherapy advances in the treatment of diffuse large B-cell lymphoma (DLBCL) patients. The multiple clinical trials and the follow-up of DLBCL patients reflect the complexity and the heterogeneity of this or these disease (s). The manuscript covers several different aspects and is well organized.

Nevertheless, the authors should discussed the potential link between the efficiency of all these treatments including novel anti-CD20 antibodies and the heterogeneity of this entity of non-Hodgkin lymphoma called DLBCL. In addition, the authors must emphasize the importance of developing biological biomarkers for personalized and precision treatment of these patients using novel anti-CD20 antibodies or new immunotherapy approaches.

The authors should focus on the most relevant clinical trials allowing more clarity of paper and the simplicity of reading.

Some paragraphs need to be merged and the tables can be simplified.

Author Response

Response to Reviewer 2 Comments

Papageorgiou et al described novel anti-CD20 antibodies and new immunotherapy advances in the treatment of diffuse large B-cell lymphoma (DLBCL) patients. The multiple clinical trials and the follow-up of DLBCL patients reflect the complexity and the heterogeneity of this or these disease (s). The manuscript covers several different aspects and is well organized.

We would like to thank the Reviewer for the valuable and most encouraging comments.

Nevertheless, the authors should discussed the potential link between the efficiency of all these treatments including novel anti-CD20 antibodies and the heterogeneity of this entity of non-Hodgkin lymphoma called DLBCL. In addition, the authors must emphasize the importance of developing biological biomarkers for personalized and precision treatment of these patients using novel anti-CD20 antibodies or new immunotherapy approaches.

We agree that DLBCL represents a heterogeneous entity. This underlying heterogeneity might affect the efficiency of the novel agents. Moreover, as the therapeutic armamentarium of DLBCL expands, the need for biomarker that could predict the response to those treatment is increasing. We discuss this in the ultimate paragraph of the revised manuscript.

The authors should focus on the most relevant clinical trials allowing more clarity of paper and the simplicity of reading.

In the revised manuscript we eliminated references on small preliminary trial that might confuse the reader. We believe that the revised version, remaining comprehensive is more easily-read.

Some paragraphs need to be merged and the tables can be simplified.

Following also the comments of Reviewer 1, we eliminated the section on CAR T cells. Table 6 has been eliminated from the revised manuscript as well as all the Tables referring to CAR T cells, therefore the revised manuscript is more concise and focused.

Reviewer 3 Report

On line 29 probably is better to specify that not yet approved in all Countries.

No other comments, is a really interesting and comprehensive review.

Author Response

Response to Reviewer 3 Comments

On line 29 probably is better to specify that not yet approved in all Countries.

No other comments, is a really interesting and comprehensive review.

We thank the Reviewer for the encouraging comments. The abstract has been revised as to emphasize that currently only PolaV is available in Europe, whereas loncastuximab tesirine and tafasitamab are only available in the US.

Round 2

Reviewer 2 Report

the manuscript can be accepted